# Causes and Treatment of Hypoxia during Total Hip Arthroplasty in Elderly Patients: A Case Report

**DOI:** 10.3390/ijerph182412931

**Published:** 2021-12-08

**Authors:** Jae Young Ji, Jin Hun Chung, Nan Seol Kim, Yong Han Seo, Ho Soon Jung, Hea Rim Chun, Hyung Yoon Gong, Woo Jong Kim, Jae Min Ahn, Yu Jun Park

**Affiliations:** 1Department of Anesthesiology and Pain Medicine, Soonchunhyang University Hospital Cheonan, 31, Suncheonhyang 6-gil, Dongam-gu, Cheonan 31151, Korea; anesth70@schmc.ac.kr (J.H.C.); nskim1977@schmc.ac.kr (N.S.K.); c75501@schmc.ac.kr (Y.H.S.); dyflam@schmc.ac.kr (H.S.J.); blau00@schmc.ac.kr (H.R.C.); 83466@schmc.ac.kr (H.Y.G.); koc_6969@naver.com (Y.J.P.); 2Department of Orthopaedic Surgery, Soonchunhyang University Hospital Cheonan, 31, Suncheonhyang 6-gil, Dongam-gu, Cheonan 31151, Korea; kwj9383@hanmail.net; 3Department of Neurosurgery, Soonchunhyang University Hospital Cheonan, 31, Suncheonhyang 6-gil, Dongam-gu, Cheonan 31151, Korea; jmstarry21@naver.com

**Keywords:** anesthesia, atelectasis, hypoxia, elderly, total hip arthroplasty, respiration

## Abstract

Intraoperative hypoxia occurs in approximately 6.8% of surgeries and requires appropriate management to avoid poor outcomes, such as increased mortality or extended hospitalization. Hypoxia can be caused by a variety of factors, including laryngospasm, inhalational anesthetics, and surgery for abdominal pathology or hip fractures. In particular, elderly patients are more vulnerable to hypoxia due to their existing lung diseases or respiratory muscle weakness. This study presents the cases of two elderly patients who developed hypoxia during total hip arthroplasty under general anesthesia. Positive end expiratory pressure, the recruitment maneuver, and increased fraction of inspired oxygen improved hypoxia only temporarily, and patients’ oxygen saturation level again dropped to 79–80%. We suspected that hypoxia was caused by atelectasis and, therefore, resumed spontaneous respiration. Thereafter, both the patients showed an improvement in hypoxia. Intraoperative hypoxia that is suspected to be caused by atelectasis can be improved by securing sufficient lung volume for respiration through increased muscle tone with spontaneous respiration.

## 1. Introduction

Respiratory functions deteriorate with aging [1], and general anesthesia, particularly with mechanical ventilation, increases the possibility of respiratory malfunction during surgery, thereby elevating the risk for hypoxia in elderly patients [2,3]. Atelectasis is one of the many causes of hypoxia that commonly occurs in patients receiving general anesthesia [4]; it is an important condition that must not be overlooked because hypoxia caused by impaired gas exchange due to atelectasis can prolong postoperative pulmonary complications. Here, we report two cases of elderly patients with a sudden onset of hypoxia during total hip arthroplasty under general anesthesia, and describe how their condition improved upon inducing spontaneous respiration and how the surgery could thus proceed under inhalation anesthesia.

## 2. Case Presentation

**Case 1**: The first patient was a 79-year-old female individual with a history of hypertension, heart failure, and middle cerebral artery infarction. Blood pressure control and cardiac function were in good condition before surgery, and no neurological complications were observed. The patient’s pulmonary function test result was normal, although her chest X-ray revealed pneumonia in the right middle lobe, for which she had been treated. The patient underwent total hip arthroplasty under general anesthesia. Before the general anesthesia, monitoring using several modalities was instituted, including electrocardiography, a noninvasive blood pressure monitor, pulse oximeter, and bispectral index (BIS) monitor. The BIS was maintained at 40–60. Anesthesia was induced with propofol (2 mg/kg) and rocuronium (0.8 mg/kg), and intra-arterial cannulation was performed for continuous blood pressure monitoring. Approximately 20 min into the surgery, the patient’s oxygen (O_2_) saturation level dropped from 93.1% to 83.1%. While being ventilated at a fraction of inspired oxygen (FiO_2_) of 0.4, her arterial blood gas showed that the partial pressure of oxygen (PaO_2_) dropped from 161.6 to 51.2. We increased the positive end expiratory pressure (PEEP) to 10 cm H_2_O and FiO_2_ to 1.0 and performed a recruitment maneuver; however, her O_2_ saturation level increased only temporarily and dropped again to 81%. Upon suspecting atelectasis due to a collapsed lung, we reversed muscle relaxation and induced spontaneous respiration. The O_2_ saturation level recovered to 90%, and we continued the surgery with spontaneous respiration. After surgery, the patient’s O_2_ saturation level recovered to the preoperative state of 98%.

**Case 2**: The second patient was an 89-year-old male individual with a history of hypertension and delirium. Before surgery, his blood pressure was well controlled, and although he was taking dementia medicine, the patient was able to follow commands well. His pulmonary function test results indicated an obstructive pattern. Total hip arthroplasty was performed using the same anesthetic regimen used for the first patient. While ventilating at an FiO_2_ of 0.4, the patient showed an onset of hypoxia, with O_2_ saturation level dropping from 100% to 80% and PaO_2_ dropping from 129 to 53.0. This patient also showed an improvement of O_2_ saturation level from 81% to 88% after recovering spontaneous respiration by administering a muscle relaxant-reversing agent. His O_2_ saturation level improved to 90% with continuous positive airway pressure. Similar to the first patient, the second patient’s O_2_ saturation level improved to 98% after surgery.

Neither patient developed any respiratory complications after surgery. The first patient had no notable findings on the postoperative chest X-ray, whereas the second patient showed subsegmental atelectasis on the right middle lobe compared with the preoperative findings (Figure 1).

## 3. Discussion

The incidence of intraoperative hypoxia is approximately 6.8%, and it is known to commonly affect patients undergoing surgery for abdominal pathology or hip fractures [5]. In contrast, hypoxia after hip fracture surgery is reportedly associated with delirium, general anesthesia, and depression [6].

Other causes of intraoperative and postoperative hypoxia include old age, obesity, American Society of Anesthesiologists status, and operation duration. Hypoxia also commonly occurs in older patients receiving general anesthesia [5].

The elastic recoil force and vital capacity of the lungs decrease with age, as does chest wall expansion. Furthermore, weakening of the respiratory muscles increases the risk of hypoxia and other respiratory complications [2]. Although mechanical ventilators assist respiration in patients under general anesthesia and those with diminished respiratory functions, it can trigger volutrauma, atelectrauma, and lung inflammation [3]. Moreover, various factors, such as loss of intercostal muscle function and surgical manipulation during general anesthesia, provoke and exacerbate atelectasis [7]. Lung computed tomography (CT) in the supine position can show normal findings during spontaneous respiration, even when atelectasis can be observed under general anesthesia (Figure 2).

It is argued that recovery from atelectasis occurs within 24 h of surgery and that no special preventive measures are required [9]. However, leaving atelectasis untreated may allow hypoxia to persist or the development of pneumonia after surgery; therefore, atelectasis must not be neglected [10]. Atelectasis can induce systemic hypoxia or lung inflammation by causing collapse of the alveoli (Figure 3) [3,4]. Hypoxia can be resolved by increasing the FiO_2_ or tidal volume, monitoring the tube position, and applying PEEP. In addition, shifting from inhalational to intravenous anesthetics (propofol) can help improve hypoxia [11]. In our cases, all of the aforementioned measures were attempted (except for switching to propofol); however, the hypoxia was only temporarily improved and ultimately recurred. In our patients, it was thought that the hypoxia occurred due to the development of atelectasis subsequent to the collapse of pulmonary alveolus due to the decrease in muscle tone. Therefore, hypoxia improved when we—rather boldly—reversed muscle relaxation during surgery and maintained spontaneous respiration under inhalation anesthesia.

We suspected that the hypoxia improved as the cephalad displacement of the diaphragm caused by muscle relaxation was reversed. This occurred as the patients’ muscle tone improved through administration of a muscle relaxant-reversing agent [4], which led to inflation of the lungs and a reduction in atelectasis. Thus, conversion to spontaneous breathing may be effective for treating perioperative hypoxia that does not respond to other treatment modalities. The second patient showed a further improvement in hypoxia with continuous positive airway pressure (CPAP) along with conversion to spontaneous respiration. CPAP appears to have facilitated ventilation and gas exchange, presumably by re-expanding the collapsed alveoli [12].

One limitation of this case report is that we cannot definitely confirm that hypoxia was caused by atelectasis, as imaging studies, such as chest X-ray or lung CT, cannot be performed after a sudden onset of hypoxia during surgery. Furthermore, the second patient did not show any abnormal findings after surgery, other than mild atelectasis in the right lower lobe, compared with their preoperative condition.

However, although we had suspected perioperative pulmonary aspiration due to old age and use of the decubitus position [5], we were able to rule out hypoxia caused by aspiration based on the absence of secretions in the tracheal suction, crackling sounds on auscultation, and findings of aspiration pneumonia on postoperative chest X-ray. We also considered the possibility of a bronchospasm; however, this was ruled out based on the lack of a wheezing sound on auscultation [13] and improvement of the hypoxia solely through spontaneous breathing, without a bronchodilator.

We also considered the possibility of pulmonary embolism in relation to the hip fracture [14], but we were able to rule it out as the hypoxia symptoms resolved after full recovery of spontaneous respiration postoperatively. In addition, there were no symptoms suggestive of pulmonary embolism, such as bloody sputum and elevated heart rate.

After conversion to spontaneous respiration, there were no notable movements at the surgical site until conclusion of the surgery. This is presumably attributable to the fact that elderly patients are more sensitive to anesthetics with a monitored anesthesia care rate of 1.48%, compared with the 2.49% in children and 1.71–2.056% in adults [15]. Furthermore, inhalational anesthetics inhibit motor responses to harmful stimuli [16]. However, additional studies on muscle relaxation caused by inhalational anesthetics are needed due to the lack of data on the use of inhalational anesthetics without muscle relaxants.

## 4. Conclusions

As intraoperative hypoxia in elderly patients can lead to postoperative pulmonary complications, thereby increasing mortality or prolonging hospitalization, continued research is needed on its various causes and the treatment of its symptoms. As described in our cases, clinicians should note the possibility of hypoxia caused by an interaction between atelectasis, resulting from the use of muscle relaxants for general anesthesia, and a patient’s pre-existing pulmonary disease. Restoring spontaneous respiration under anesthesia can be a treatment option for hypoxia. 

## Figures and Tables

**Figure 1 ijerph-18-12931-f001:**
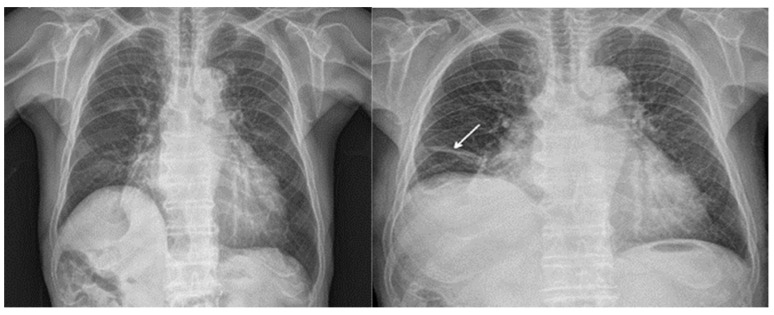
Atelectasis had formed in the right lower lobe (arrow) of the second patient’s lung postoperatively (**right**), which was not present preoperatively (**left**).

**Figure 2 ijerph-18-12931-f002:**
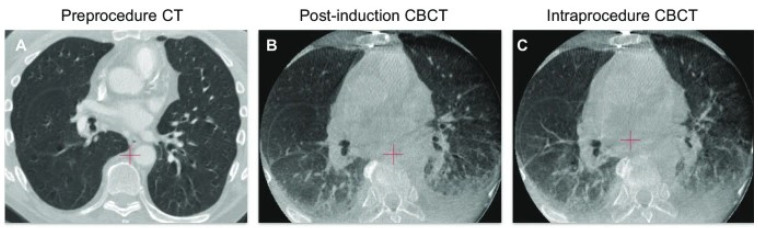
(**A**) A patient’s lungs before anesthesia induction. (**B**,**C**) Atelectasis in the dependent regions of both lungs after anesthesia induction. Cone beam CT (CBCT) showing exacerbated atelectasis during surgery [8].

**Figure 3 ijerph-18-12931-f003:**
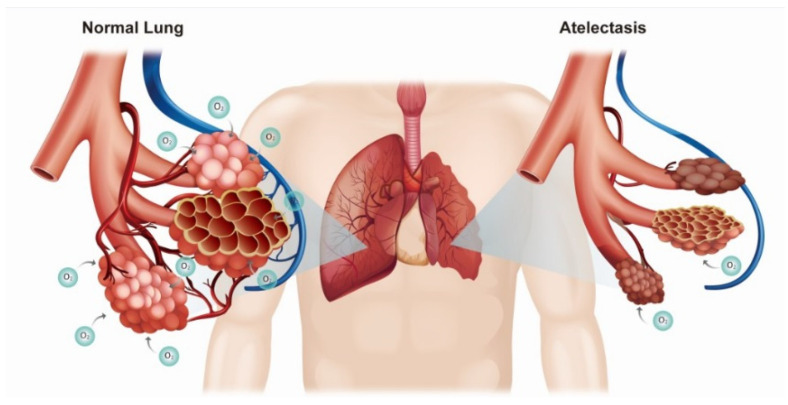
Diagram showing ineffective gas exchange in a left lung due to atelectasis compared with a normal right lung.

## Data Availability

Data sharing is not applicable to this article as no datasets were generated or analyzed during the current study.

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
