# Peer review of "Causes and Treatment of Hypoxia during Total Hip Arthroplasty in Elderly Patients: A Case Report"

_ijerph, 2021, doi:10.3390/ijerph182412931_

Round 1
Reviewer 1 Report
The reviewed study presents the cases of two elderly patients who developed hypoxia during total hip arthroplasty under general anesthesia.
The manuscript is well structured, with the primary purpose so clear. The data partly support conclusions. The results highlighted that atelectasis due to general anesthesia can cause hypoxia during surgery and pulmonary complications after surgery and various measures should be considered to improve hypoxia. If hypoxia is not resolved using the treatment modalities reported previously, resuming spontaneous respiration can be a viable alternative to treat hypoxia. I have several suggestions:
- Introduction:
Would you please give examples of sudden onset of hypoxia during total hip arthroplasty under general anesthesia? Has this topic already been discussed in the literature? Please discuss this issue in more detail in the introduction section.
- Conclusions
Would you please provide more detailed conclusions concerning the discussed clinical cases
Overall:
Please standardize the citation system according to the journal's requirements.
Author Response
Response to Reviewer 1’s comments
- Introduction:
Would you please give examples of sudden onset of hypoxia during total hip arthroplasty under general anesthesia?.
Has this topic already been discussed in the literature? Please discuss this issue in more detail in the introduction section.
Response: There are no reports of hypoxia occurring during total hip replacement; however, cases of postoperative hypoxia due to delirium or a reaction to the anesthesia have been reported. Intraoperative hypoxia occurs in about 6.8% of patients. We have added the relevant information to the revised Introduction and Discussion.
- Conclusions Would you please provide more detailed conclusions concerning the discussed clinical cases
Response: As intraoperative hypoxia in elderly patients can increase the rates of mortality or postoperative pulmonary complications, thereby prolonging hospitalization, continued research into its various causes as well as the treatment of its symptoms is warranted. As described in our cases, clinicians should note the possibility of hypoxia caused by an interaction between atelectasis, subsequent to the use of muscle relaxants for general anesthesia, and a patient’s pre-existing pulmonary disease. Restoring spontaneous respiration under anesthesia can be a treatment option for hypoxia.
Overall:
Please standardize the citation system according to the journal's requirements.
Response: Thank you for pointing this out. We have revised the in-text citations and list of references to ensure compliance with journal requirements.

Reviewer 2 Report
Excellent cases of complications are quite frequent in this population group. I think they are sufficiently described, and with adequate image support. However, I believe that other possible causes, such as thromboembolism, should be ruled out. On the other hand, the recruitment maneuvers and the increase in FiO2 do not produce a significant increase in O2 sat, which leads us to think that there are other causes involved.
Author Response
1.Excellent cases of complications are quite frequent in this population group. I think they are sufficiently described, and with adequate image support. However, I believe that other possible causes, such as thromboembolism, should be ruled out. On the other hand, the recruitment maneuvers and the increase in FiO2 do not produce a significant increase in O2 sat, which leads us to think that there are other causes involved.
Response: We have described how we considered the possibility of pulmonary embolism associated with total hip arthroplasty in the revised Discussion. We ultimately determined that it was unlikely because the hypoxia improved with spontaneous respiration, and the patient did not show symptoms of pulmonary embolism postoperatively.
Although other causes cannot be completely ruled out, we suspected post-anesthetic atelectasis based on the fact that the hypoxia, which had not been present before inducing anesthesia, was improved solely by restoring spontaneous respiration amid a rapid drop in saturation. We believe that the hypoxia was exacerbated by the patient’s advanced age and pre-existing pulmonary disease.

Reviewer 3 Report
Great job.
I would like to comment to the authors that the last two terms used in key words "old age and Spontaneous respiration" do not appear as such, in the MesH descriptors. It is recommended to change for others and add a couple more (up to six in total) so that potential readers can more easily find the article and increase its potential visibility and therefore impact.
Discussion:
Have you considered the option of submitting potential patients who may develop this hypoxia to an evaluation test prior to the intervention (for example, spirometry) and a respiratory physiotherapy also prior to surgery that can increase their lung volumes? (tidal volume, vital capacity, etc)
Patients who are going to undergo thoracic and / or abdominal surgeries usually undergo this type of prior training to minimize the consequences of anesthesia and / or surgery in a ventilatory pattern.
Author Response
Great job.
I would like to comment to the authors that the last two terms used in key words "old age and Spontaneous respiration" do not appear as such, in the MesH descriptors. It is recommended to change for others and add a couple more (up to six in total) so that potential readers can more easily find the article and increase its potential visibility and therefore impact.
Response: We have revised and supplemented original list of key words accordingly. The final list now comprises 6 Mesh terms.
Discussion:
Have you considered the option of submitting potential patients who may develop this hypoxia to an evaluation test prior to the intervention (for example, spirometry) and a respiratory physiotherapy also prior to surgery that can increase their lung volumes? (tidal volume, vital capacity, etc)Patients who are going to undergo thoracic and / or abdominal surgeries usually undergo this type of prior training to minimize the consequences of anesthesia and / or surgery in a ventilatory pattern.
Response: We are not sure whether we have understood the question correctly, but unlike during surgeries of the upper and lower extremities, spontaneous breathing can hinder abdominal and chest procedures. Hence, continuous positive end expiratory pressure or a periodic recruitment maneuver can be considered initial options for improving hypoxia in such cases. If the hypoxia persists, we may restore muscle tone and attempt to recover spontaneous breathing after discussion with the surgeon.

Round 2
Reviewer 1 Report
Dear Authors,
Thank you for considering my suggestions. I hope my comments have helped to improve the quality of your work. In its current form, the article meets the criteria for publication in the IJERPH journal.